# Membrane-Associated, Not Cytoplasmic or Nuclear, FGFR1 Induces Neuronal Differentiation

**DOI:** 10.3390/cells8030243

**Published:** 2019-03-14

**Authors:** Katalin Csanaky, Michael W. Hess, Lars Klimaschewski

**Affiliations:** 1Division of Neuroanatomy, Medical University of Innsbruck, 6020 Innsbruck, Austria; katalin.csanaky@i-med.ac.at; 2Division of Histology and Embryology, Medical University of Innsbruck, 6020 Innsbruck, Austria; michael.hess@i-med.ac.at

**Keywords:** optogenetics, FGF2, ERK, AKT, receptor kinase, neurite outgrowth, HEK293, PC12

## Abstract

The intracellular transport of receptor tyrosine kinases results in the differential activation of various signaling pathways. In this study, optogenetic stimulation of fibroblast growth factor receptor type 1 (FGFR1) was performed to study the effects of subcellular targeting of receptor kinases on signaling and neurite outgrowth. The catalytic domain of FGFR1 fused to the algal light-oxygen-voltage-sensing (LOV) domain was directed to different cellular compartments (plasma membrane, cytoplasm and nucleus) in human embryonic kidney (HEK293) and pheochromocytoma (PC12) cells. Blue light stimulation elevated the pERK and pPLCγ1 levels in membrane-opto-FGFR1-transfected cells similarly to ligand-induced receptor activation; however, no changes in pAKT levels were observed. PC12 cells transfected with membrane-opto-FGFR1 exhibited significantly longer neurites after light stimulation than after growth factor treatment, and significantly more neurites extended from their cell bodies. The activation of cytoplasmic FGFR1 kinase enhanced ERK signaling in HEK293 cells but not in PC12 cells and did not induce neuronal differentiation. The stimulation of FGFR1 kinase in the nucleus also did not result in signaling changes or neurite outgrowth. We conclude that FGFR1 kinase needs to be associated with membranes to induce the differentiation of PC12 cells mainly via ERK activation.

## 1. Introduction

The fibroblast growth factor receptor (FGFR) family comprises four closely related receptors (FGFR1-4) consisting of a signal peptide, three extracellular immunoglobulin (Ig)-like domains, an acidic box, a single transmembrane helix, and an intracellular split tyrosine kinase domain [1]. The binding of FGF ligands results in receptor dimerization and autophosphorylation of the cytoplasmic kinase domains. Following the recruitment of various adapter molecules, several intracellular signaling pathways are activated, of which, the Ras/extracellular signal-regulated kinase (ERK) and phosphatidylinositol-3 kinase (PI3K)/AKT signaling cascades play a key role in neuronal differentiation and axon growth [2]. All FGFR subtypes drive the same signaling cascades but with different strengths [3].

Activated FGFR complexes are endocytosed and partly recycled to the plasma membrane or transported to late endosomes/multivesicular bodies followed by degradation in lysosomes [4]. The internalization of activated FGFR1 does not implicate the inactivation of the receptor or attenuation of signaling, but allows ongoing and even stronger signaling activities particularly of the ERK pathway. In neurons, ERK1/2 activation requires receptor internalization, while AKT activation does not [5]. We and others have demonstrated that the extent of the stimulation of the major signaling pathways involved in neuronal differentiation and axon outgrowth is crucially dependent on the amount and localization of activated kinase domains. For example, leupeptin, a tripeptide inhibitor of cysteine and aspartic acid cleaving proteases, prevents the lysosomal degradation of FGFR1 and promotes basic fibroblast growth factor (FGF2)-induced axon regeneration by enhanced receptor recycling [6].

In addition to its membrane localization, FGFR1 is constitutively found in the cytoplasm and nucleus [7]. The nuclear translocation of FGFR1 is importin-dependent [8] and results in the binding of the receptor to transcriptionally active chromatin which drives the gene expression of FGF2 and tyrosine hydroxylase [9]. Furthermore, interaction with RSK1 promotes FGFR1 release from the pre-Golgi compartments to the cytoplasm, increases the mobile population of FGFR1 and facilitates the nuclear accumulation of FGFR1 [10,11]. Nerve growth factor (NGF) has also been suggested to utilize integrative nuclear FGFR1 signaling (INFS) for its gene-activating functions [12]. In addition, INFS is apparently involved in the dendritic outgrowth of sympathetic neurons treated with bone morphogenetic protein [13].

The plethora of FGFR-dependent biological effects is not only explained by differences in subcellular receptor targeting but also by the differential expression of adapter proteins and other signaling components. Since the duration and kinetics of receptor activation play a decisive role in determining the functional and morphological outcome [14], FGFR1 represents an ideal object for optogenetic manipulation.

Recently, light-activatable FGF receptors that are incapable of ligand binding were developed [15]. Chimeric opto-fibroblast growth factor receptors (opto-FGFRs) exhibiting the catalytic domain were fused to the algal light-oxygen-voltage-sensing (LOV) domain for light-induced dimerization. These constructs lack the extracellular receptor domain and are, therefore, insensitive to endogenous or exogenous ligands. Short pulses of blue light induce LOV domain dimerization which results in the transphosphorylation of two receptor kinase molecules. Receptor phosphorylation diminishes rapidly after light stimulation and is followed by the inactivation of the main signaling pathways. Thus, intracellular FGFR activity can be externally controlled by repeated light stimulation. In this study, plasma membrane-targeted (mem-opto-FGFR1), cytoplasmic (cyto-opto-FGFR1) and nuclear FGFR1 (nucl-opto-FGFR1) were produced and studied with regard to signal pathway activation and neurite outgrowth.

## 2. Materials and Methods

### 2.1. Plasmid Construction

Mem-opto-FGFR1, which is in the pcDNA3.1(−) plasmid backbone, was described previously [15]. To clone cyto-opto-FGFR1, the myristoylation signal (MYR) of mem-opto-FGFR1 was replaced by the Kozak sequence using reverse PCR. Three nuclear localization sequences (NLSs) were inserted into cyto-opto-FGFR1 to obtain nucl-opto-FGFR1. Fluorescently labeled versions of opto-FGFR1 were produced by inserting mVenus (mV) (Figure 1). All plasmids that were generated were verified by sequencing.

### 2.2. Cell Culture and Transfection

Human embryonic kidney cells (HEK293, ATCC, Manassas, VA, USA) were grown in Dulbecco’s Modified Eagle’s Medium (DMEM, Sigma, St. Louis, MI, USA) supplemented with 10% fetal bovine serum (FBS; Thermo Fisher Scientific, Waltham, MA, USA, TFS) and 1% antibiotic–antimycotic solution (100 U/mL penicillin, 100 μg/mL streptomycin, 0.25 μg/mL amphotericin, TFS). Rat adrenal pheochromocytoma (PC12) cells (Sigma) were grown in RPMI (Roswell Park Memorial Institute) 1640 (TFS) supplemented with 10% horse serum, 5% FBS, and 1% antibiotic–antimycotic solution on dishes coated with 50 µg/mL collagen type I (Merck, Kenilworth, NJ, USA). Human glioblastoma cells (U251) were cultivated in RPMI 1640 cell culture medium supplemented with 5% FBS and 1% antibiotic–antimycotic solution. All cell lines were grown under standard conditions in humidified atmosphere at 37 °C, with 5% CO_2_. Transient transfections were performed with Lipofectamine 2000 (Invitrogen, Carlsbad, CA, USA) according to the manufacturer’s protocol.

### 2.3. Confocal Laser Scanning Microscopy

For the localization of mV-opto-FGFR1s in HEK293 cells, 4 × 10^5^ cells were seeded on #1.5 coverslips and co-transfected with 1 µg mV-opto-FGFR1 and 1 µg LifeAct–mCherry (Addgene #40908). Six hours after transfection, 25 µM importazole (Sigma) was added to the cells overnight to prevent nuclear import of the mV-nucl-opto-FGFR1 construct. Cells were fixed in 4% formaldehyde solution (made from paraformaldehyde; PFA), stained with Hoechst 33258 dye, mounted in ProLong Diamond Antifade Mountant (TFS) and imaged using SP8 confocal laser scanning microscopy (Leica, Wetzlar, Germany).

### 2.4. Immunogold Electron Microscopy

For immunogold electron microscopy, Tokuyasu cryosection and cryo-based pre-embedding immunogold labelling were applied [16,17,18]. HEK293 cells were transfected with 1 µg mV-mem-opto-FGFR1 in 10-cm diameter dishes and kept overnight in DMEM supplemented with 10% FBS and 1% antibiotic–antimycotic solution. Cells were fixed with 4% (*w*/*v*) buffered formaldehyde solution for >3 days at room temperature (RT). Ultra-thin cryosections were labelled with goat anti-GFP (#600-101-215, 1:500, Rockland, Limerick, PA, USA), followed by rabbit anti-goat IgG 15-nm colloidal gold conjugates (#EM.RAG15, 1:50, British Biocell Intl., Cardiff, UK). U251 cells were cultured on sapphire coverslips (Ø 3 mm), transfected with 0.2 µg mV-mem-opto-FGFR1 and kept overnight in RPMI 1640 supplemented with 5% FBS and 1% antibiotic–antimycotic solution. Samples were then subjected to high-pressure freezing, freeze-substitution, and rehydration, followed by pre-embedding labelling. Primary antibodies were goat anti-GFP (#600-101-215, 1:2000, Rockland), secondary antibodies were Nanogold^®^-Fab’ rabbit anti-goat IgG (H+L) (#2006; 1:700), followed by silver enhancement (SE) with HQ-Silver^®^ (#2012, all from Nanoprobes) and standard plastic embedding. Sections were analyzed by transmission electron microscopy.

### 2.5. Stimulation and Immunoblot

HEK293 cells were seeded on 6-well dishes (1 × 10^6^ per dish, Corning, NY, USA) and transfected with either 2 µg opto-FGFR1 or 2 µg FGFR1–eGFP [19]. Six hours after transfection, the medium was replaced with reduced serum medium (0.5% FBS) and the cells were incubated overnight. One day after transfection, opto-FGFR1-transfected cells were stimulated for 5 min by 2.5 µW/mm^2^ blue light in a reptile egg incubator (PT2499, Exo Terra, Hagen, Holm, Germany) equipped with 300 light-emitting diodes. The blue light intensity was measured with a digital power meter connected to the Microscope Power Sensor Head (Thorlabs, Newton, NJ, USA). Cells that were kept in the dark (wrapped in aluminum foil) in the same incubator served as controls. FGFR1–eGFP-transfected cells were stimulated by treating them with 100 ng/mL FGF2 (Stemcell, Vancouver, BC, Canada) and 1 µg/mL heparin sulfate (HepS, Sigma) for 5 min.

The lysis of light-stimulated cells was performed in the incubator under blue light. The dark samples were lysed under non-stimulating red light. Cells were harvested in 50 µL lysis buffer on ice, sonicated and centrifuged (13,000 rpm, 20 min, 4 °C). To prevent ERK signaling, 50 µM MAPK kinase inhibitor (PD98059; Sigma) was added to the cells 2 h prior to light activation or FGF2 stimulation. For the measurement of ERK kinetics, the ligand was washed out two times with PBS and lysates made after 5, 15 and 30 min. In the case of opto-FGFR1-transfected cells, the lysates were obtained after 5, 15 and 30 min of dark period. A total of 30 µg protein per lane was separated by SDS-PAGE and transferred to Immobilon-FL PVDF membranes (Merck, Kenilworth, NJ, USA). Blots were incubated with primary antibodies in Odyssey Blocking Buffer (LI-COR) overnight at 4 °C (pFGFR1, #3476, 1:1000; ERK1/2, #9107, 1:1000; pERK1/2, #9101, 1:2000; AKT, #2920, 1:1000; pAKT, #4060, 1:2000; pPLCγ1, #2821, 1:2000, all from Cell Signaling Technology/CST, Danvers, MA, USA; PLCγ1, #16955, 1:1000, Abcam; GAPDH, sc-32233, 1:500, Santa Cruz Biotechnology, Dallas, TX, USA). Secondary antibodies (IRDye 800CW goat anti-rabbit IgG, #926-32231, and IRDye 680RD goat anti-mouse IgG, #926-68070, both 1:20.000, LI-COR, Lincoln, NE, USA) were applied for 1 h at RT. Signals were recorded with an infrared fluorescence detection system (Odyssey Fc Imaging System).

Long-term light and ligand stimulation for neurite outgrowth assay and immunocytochemistry pheochromocytoma (PC12) cells was used for neuronal differentiation experiments. Cell lines enriched with mV-mem-opto-FGFR1-, mV-cyto-opto-FGFR1- or mV-nucl-opto-FGFR1-transfected cells were obtained as follows. Cells with a low passage number (<P10) were seeded on 6-well plates (BD) and transiently transfected with 1 µg of plasmids. After 24 h, the culture medium was replaced by a medium containing 300 µg/mL Geneticin (G418, TFS), which was changed every other day. When sufficient numbers of resistant cell colonies were observed, the cells were triturated with fire-polished Pasteur pipettes for 10 min and then transferred to collagen-coated culture dishes. For microscopic analysis, cells were plated on 35-mm diameter collagen-coated imaging dishes (IBIDI). PC12 cells were also transiently transfected with 1 µg FGFR1–eGFP. After 6 h, cells were cultured in RPMI 1640 supplemented with 10% horse serum, 5% FBS, and 1% antibiotic–antimycotic solution overnight followed by a starvation medium supplemented with 2 mM L-glutamin, 1% antibiotic–antimycotic solution, and N2 supplement (100×, TFS) for 2 h. FGFR1–eGFP-transfected cells and naive PC12 cells were then treated for 48 h with 100 ng/mL FGF2 + 1 µg/mL HepS and with 100 ng/mL nerve growth factor (NGF), respectively. For light stimulation, the dishes were placed in a humidified atmosphere at 37 °C with 5% CO_2_ in an incubator with light-emitting diodes as described above. The blue light intensity was measured with a digital power meter and set to 2.5 μW/mm^2^. The control cells were kept in the dark in the same incubator during stimulation. A repetitive 5 min ON and 55 min OFF cycle was used for stimulation over 48 h.

Cells were then fixed with 4% PFA for 15 min and their neurites were visualized by immunocytochemistry. Neuron-specific class III beta-tubulin (1:1000, 1 h at RT; R&D system) and Alexa 568 goat anti-mouse IgG (1:1000, 2 h at RT; TFS) were applied as the primary and secondary antibody, respectively. Inverted fluorescence microscopy (AxioVert, Zeiss, Oberkochen, Germany) at 40× magnification was used for imaging. Neurite outgrowth was defined if cells exhibited at least one process of more than one cell body diameter length.

For immunocytochemistry, PC12 cells enriched with mV-opto-FGFR1-transfected cells were used. Following serum starvation, light stimulation and fixation, cells were incubated overnight with primary antibodies (pERK1/2, #9101, 1:400, CST; pAKT, #4060, 1:400, CST) followed by incubation at RT for 2 h with Alexa 546 goat anti-rabbit IgG (1:1000, TFS). SP8 confocal laser scanning microscopy (Leica) was used for cell imaging. Controls included the omission of primary antibodies (negative) and stimulation of cells with NGF (positive).

### 2.6. Quantification of Results and Statistical Analysis

The total neurite length (TNL, sum of the length of all neurites extending from the cell body), maximal distance (MD, length of longest neurite) and number of neurites per cell were quantified (Appendix A). Metamorph software version 4.6 (Visitron Systems, Puchheim, Germany) was used for the quantification of neurite outgrowth and immunocytochemistry. Band intensities of the Western blots were densitometrically quantified using Image Studio LiteVer 5.2. GraphPad Prism software was used for statistical analysis (two-way ANOVA with Sidak correction).

## 3. Results

### 3.1. Localization of mV-opto-FGFR1s

In HEK293 cells, mV-mem-opto-FGFR1 localized predominantly to the plasma membrane and to endosomes (Figure 2A1–A4), while mV-cyto-opto-FGFR1 diffused freely in the cytoplasm and nucleus (Figure 2B1–B4). The mV-nucl-opto-FGFR1 construct was found exclusively in the nucleus with enrichment of the protein in presumptive nuclear speckle domains (Figure 2C1–C4). Inhibition of the importin β transport receptors with 25 µM importazole resulted in diffuse cytoplasmic yellow fluorescence of mV-nucl-opto-FGFR1, indicating a partial block of the nuclear import (Appendix A). Immunogold electron microscopy of the thawed cryosections revealed mV-mem-opto-FGFR1 in the plasma membrane, in (late) endosomes/multivesicular bodies (MVBs) and lysosomes of HEK293 cells (Figure 2D1–D3). These data were further confirmed by a complementary immunogold labeling method applied to a different cell line (U251 glioblastoma cells) in which the same subcellular distribution of mV-mem-opto-FGFR1 was found (Appendix A).

### 3.2. Opto-FGFR1-Dependent Signaling Pathway Activation in HEK293 Cells

For all quantitative signaling assays, the levels of active FGFR1 (pFGFR1) and the control protein (GAPDH) were determined to normalize for different transfection efficiencies (Figure 3A,B). Immunoblot results showed that serum starvation (naive state) significantly reduced the basal level of pERK but not the basal levels of pAKT or pPLCγ1 in HEK293 cells. The overexpression of FGFR1–eGFP induced ERK and AKT activation which was further increased by adding FGF2 for 5 min. No marked changes in pPLCγ1 levels were observed following FGF2 treatment.

The overexpression of mem-opto-FGFR1 alone (without light stimulation) resulted in receptor autoactivation, as indicated by increased pERK and pPLCγ1 but unchanged pAKT levels (Figure 3A). Autoactivation was not observed after cyto-opto-FGFR1 or nucl-opto-FGFR1 transfection. Blue light stimulation for 5 min following mem-opto-FGFR1 or cyto-opto-FGFR1 transfection elevated the pERK levels (Figure 3C). Ligand-induced ERK activation lasted longer, whereas light-induced ERK activation was terminated faster (30 min versus 5–15 min, Appendix A). Minimal ERK activation was observed after transfection with nucl-opto-FGFR1. Unexpectedly, the pAKT levels remained unchanged after blue light stimulation of HEK293 cells following transfection with any of the opto-FGFR1 constructs (Figure 3D). On the other hand, the pPLCγ1/tPLCγ1 ratio slightly increased after blue light stimulation of mem-opto-FGFR1- but not of cyto- or nucl-opto-FGFR1-transfected cells (Figure 3E).

### 3.3. Immunocytochemistry of mV-opto-FGFR1-Transfected PC12 Cells

Immunocytochemistry was performed instead of protein blotting to visualize signaling pathway activation in PC12 cells (due to their lower transfection rate). Similarly to HEK293 cells, mem-opto-FGFR1-transfected pheochromocytoma cells exhibited increased pERK levels, as indicated by intense pERK immunolabeling in the cytoplasm (Figure 4A2/A6,G). As with HEK293 cells, no AKT activation was observed following intermittent blue light stimulation (Figure 4B2/B6,H). Furthermore, the pERK and pAKT levels were unchanged in stimulated mV-cyto-opto-FGFR1 (Figure 4C2/C6,D2/D6) or mV-nucl-opto-FGFR1-expressing PC12 cells (Figure 4E2/E6,F2/F6).

### 3.4. Neuronal Differentiation of PC12 Cells Induced by Blue Light

PC12 cells exhibited no spontaneous or FGF2-induced neurite outgrowth, suggesting that the clone used in the present study does not express significant levels of endogenous FGF receptors (Figure 5A and Appendix A). In fact, all four FGFR mRNAs are endogenously expressed but the levels are low, particularly for FGFR1 (Appendix A). Two days after treatment with NGF, neuronal differentiation was observed (Figure 5B; 120 ± 11.9 µm total neurite length, TNL, Figure 5K; 52.7 ± 4 µm of maximal neurite length, MD, Figure 5L; 2.6 ± 0.12 processes extending from the cell body, Figure 5M). Cells transiently transfected with FGFR1–eGFP revealed significantly longer neurites compared to naive cells (Figure 5C) and increased neurite initiation (Figure 5M). FGF2 treatment further enhanced neuronal differentiation with long neurites (Figure 5D). Although the autoactivation of mV-mem-opto-FGFR1 induced mild neurite outgrowth in the dark state (Figure 5E), blue light stimulation resulted in dramatically increased neuronal differentiation (Figure 5F,K) which was significantly inhibited by prior PD98059 treatment (Appendix A). A significant increase in the number of neurites extending from mV-mem-opto-FGFR1-transfected cells after blue light stimulation was observed as well as significantly longer neurites when compared to NGF and FGF2 treatment (Figure 5L,M). Cells expressing either mV-cyto-opto-FGFR1 or mV-nucl-opto-FGFR1 showed flattened, spindle-shaped morphology with short cytoplasmic extensions but failed to grow processes longer than one cell body in diameter (Figure 5G–J).

## 4. Discussion

Light-sensitive G-protein-coupled receptors (e.g., rhodopsin) occur naturally, whereas light-sensitive receptor tyrosine kinases (RTKs) need to be artificially produced. Recent studies have been aimed at subcellular targeting of opto-TrkA and light-gated adenylate cyclase [20,21]. In addition, various membrane-associated opto-RTK constructs were synthesized, such as opto-TrkB [22] and three different opto-FGFR1 constructs [15,23,24]. One of the light-activated FGFR1 proteins (through the homointeraction of cryptochrome 2) induced cell polarization and directed cell migration through changes in the actin–tubulin cytoskeleton [23]. Furthermore, opto-FGFR1 was applied for light-induced sprouting of human bronchial epithelial cells [15].

The opto-FGFR1 constructs used here were designed for specific targeting of the kinase domain to only the plasma membrane, cytoplasm, and nucleus, respectively, to investigate the possible effects of subcellular FGFR kinase activation on signal pathway induction and neurite outgrowth as a biological read-out. Similarly to full-length FGFR1, immunoelectron microscopy revealed that mV-mem-opto-FGFR1s were anchored to the plasma membrane, internalized and transported to multivesicular bodies (MVBs)/late endosomes and lysosomes [25,26]. Although our construct was expected to only attach to membranes (plasma membrane, endosomal/lysosomal), mV-mem-opto-FGFR1 was also occasionally observed in the cytoplasm and nucleus. It is known that internalized full-length FGFR1 may be released from endosomes and travels to the nucleus through importin β-mediated translocation and that newly synthetized FGFR1 may enter the nucleus directly as well [27,28,29,30].

Intranuclear FGFR1 is localized within nuclear matrix-attached speckle domains in the form of large discrete spots [31,32,33]. In this study, such fluorescence patterns were also observed in mV-nucl-opto-FGFR1-transfected cells exhibiting the split kinase domain of FGFR1 coupled to three NLSs. Biologically active, soluble kinase fragments are also created by cleavage at the transmembrane domain [34,35]. Similarly to these natural cytoplasmic FGFR1 fragments, mV-cyto-opto-FGFR1 constructs lacking specific targeting signals diffuse freely in the cytoplasm.

In this study, the activation of ERK but not of AKT was observed in cells expressing membrane-associated or, to a lesser extent, cytoplasmic opto-FGFR1 after blue light stimulation. The light-induced activation of both pathways has been described in connection with membrane-localized opto-TrkA and -TrkB receptors and with cytoplasmic opto-TrkA [20,22]. Moreover, significant ERK phosphorylation was observed after targeting active kinase domains of FGFR1, FGFR3 and FGFR4 to the plasma membrane [36]. Our findings are consistent with these and earlier studies from our own group demonstrating that FGFR1, in contrast to TrkA, exerts a significantly stronger influence on the ERK than on the AKT pathway [37,38,39].

Opto-constructs provide additional advantages such as that activation ceases shortly after switching off the light stimulus. In contrast to ligand-dependent RTK signaling, light-induced stimulation is terminated faster. In other studies, ERK nuclear translocation and immunoblot assays indicated similar ERK phosphorylation kinetics following light-induced Raf1 and Ras recruitment, as observed in our experiments [40,41]. However, although PI3K/AKT was expected to be activated after FGFR phosphorylation, we did not observe changes in the pAKT levels after light stimulation of opto-FGFR1s in transfected HEK293 and PC12 cells. These variations in signal pathway activation between different cell lines and different modes of induction may be explained by the differential expression of signaling adapters and possible crosstalk, particularly of the ERK- and AKT-dependent pathways [42,43].

PC12 cells are widely used to study neuronal differentiation. NGF induces biochemical, electrophysiological and morphological changes in these cells, recapitulating many characteristic features of differentiated sympathetic neurons [44,45]. The optogenetic activation of Trk receptors and stimulation of the Raf–MEK–ERK cascade have also been demonstrated to induce neuronal differentiation [41]. Due to the selective activation of ERK in the latter study, the neurites were longer than those of NGF-treated cells which could be explained by multiple downstream pathway activation including PI3K/AKT-mediated process branching. Taken together, we suggest that repetitive ERK stimulation acts as the primary driver of neurite extension, whereas AKT-dependent pathways primarily stimulate the formation of branches [46].

In this study, PC12 cells transfected with mV-mem-opto-FGFR1 exhibited longer neurites than NGF- or FGF2-treated cells and significantly more neurites extended from the cell body. Interestingly, light-induced stimulation of mem-opto-FGFR1 did not increase pAKT levels as compared to dark control cells, suggesting that the effects on neurite outgrowth are mainly, if not exclusively, dependent on the ERK pathway. Our immunocytochemistry results showed intense pERK immunolabeling in the cytoplasm, but only weak fluorescence in the nucleus 55 min after blue light stimulation. In contrast to other cell lines, the activated ERK2 variant (ERK2–MEK1 fusion protein) diffusely localized to the cytoplasm and to extending cellular processes. This localization pattern suggests that ERK2 is involved in promoting neurite formation, not only through its actions on gene transcription but also through effects at the site of neurite extension [47]. Light-induced neuronal differentiation was not observed in mV-cyto- or in nucl-opto-FGFR1-transfected cells, corroborating earlier studies by Donoghue and his coworkers who achieved differentiation from plasma membrane-bound FGFR kinase domains in PC12 cells [36,48]. However, the full-length FGFR1 targeted to the nucleus also induced PC12 cell differentiation [12,49]. As discussed above, the binding of nuclear FGFR1 to CREB-binding protein (CBP) or to ribosomal S6 kinase isoform 1 (RSK1) may be involved in nuclear FGFR1 signaling [49,50]. Nevertheless, the activated kinase domain alone does not appear to be sufficient to modify gene expression, resulting in neuronal differentiation.

In summary, we demonstrated that only membrane-bound opto-FGFR1 constructs are capable of activating the ERK pathway sufficiently to induce neuronal cell differentiation in PC12 cells. We assume that fully functional signaling platforms only form in association with membranes. Active RTKs apparently recruit different adaptors and scaffold proteins in plasma membranes as compared to endosomal and other membranes which exhibit different curvatures and phosphatidylinositol compositions. Finally, the present study provides further evidence that FGFR1-dependent signaling pathways and neurite outgrowth can be controlled and manipulated optogenetically, which allows the study of subcellular receptor activation with spatial and temporal precision.

## Figures and Tables

**Figure 1 cells-08-00243-f001:**
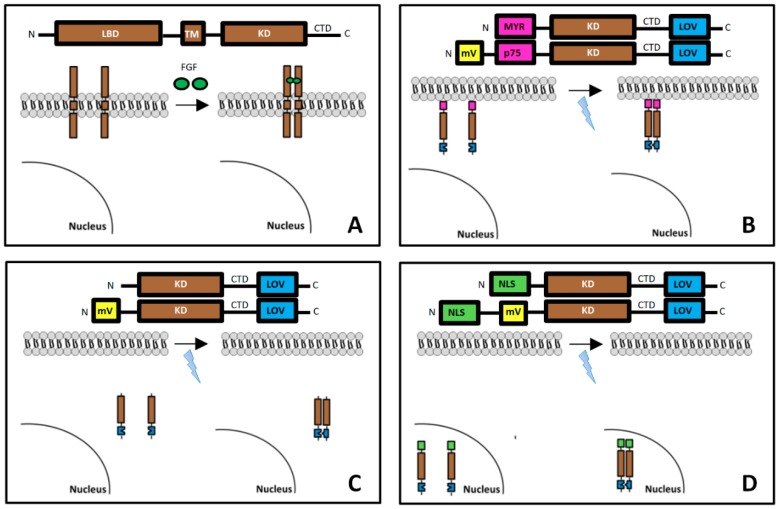
Design of light-controlled opto-fibroblast growth factor receptors (FGFR1s) and their molecular architecture: sequence of the different FGFR1 coding genes, spatial relation of FGFR1s to the surface membrane and nucleus, and their activation/dimerization. (**A**) Naturally occurring full-length FGFR1 consists of the extracellular ligand-binding (LBD), transmembrane (TM), kinase (KD) and C-terminal tail (CTD) domains. (**B**) Artificial mem-opto-FGFR1 is anchored to the plasma membrane with an N-terminal myristoylation signal (MYR) followed by the KD, CTD and LOV domain (mV-mem-opto-FGFR1 is inserted into the plasma membrane by incorporation of the transmembrane domain of p75). Fluorescent mVenus and the light-oxygen-voltage-sensing (LOV) protein are separated to avoid non-specific activation of the LOV domain by mVenus (by Förster resonance energy transfer). mVenus can be detected by excitation with green light (514 nm) that does not activate the LOV domain. (**C**) Cyto-opto-FGFR1 consists of only the KD and the LOV domain. (**D**) Three nuclear localization sequence (NLS) signals are inserted into nucl-opto-FGFR1 with or without mVenus.

**Figure 2 cells-08-00243-f002:**
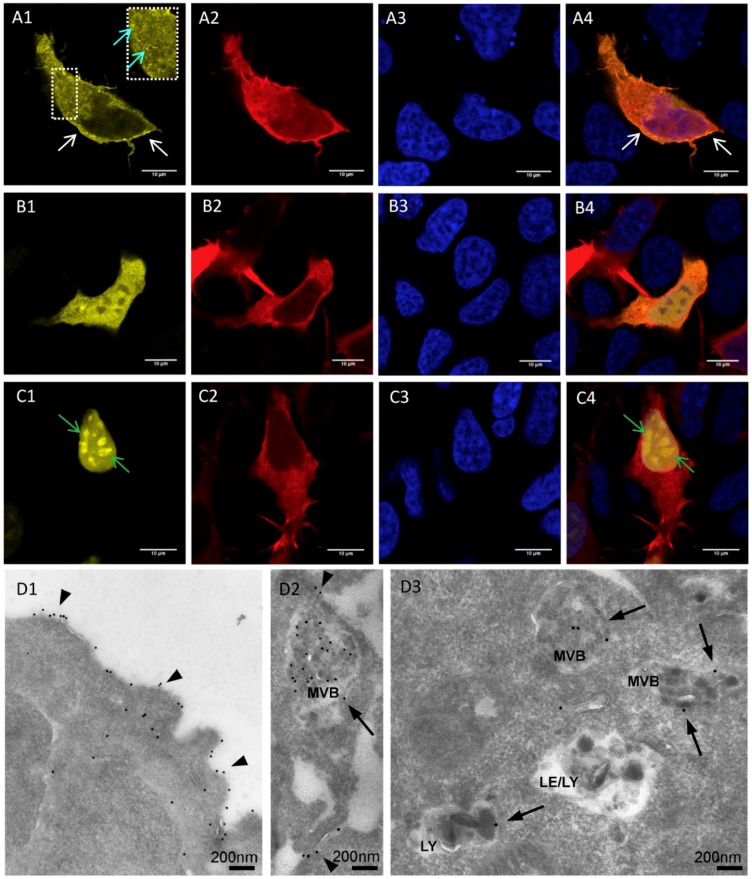
Light and electron microscopic localization of mV-opto-FGFR1s in human embryonic kidney (HEK293) cells. (**A**–**C**) Immunofluorescence microscopy of cells co-transfected with mV-opto-FGFR1s and LifeAct–mCherry to visualize cell bodies and all cytoplasmic processes. (**A1**) mV-mem-opto-FGFR1 is observed in the plasma membrane (white arrows) and in endosomes (inset; cyan arrows) indicating endosomal receptor internalization. (**B1**) mV-cyto-opto-FGFR1-transfected cells reveal diffuse yellow fluorescence in the cytoplasm and nucleus of transfected cells. (**C1**) mV-nucl-opto-FGFR1 is only located in the nucleus (nuclear speckle domains are indicated by green arrows). Fixed cell nuclei are stained with Hoechst (blue) and A4–C4 represent overlays. Scale bars for all images are 10 µm. (**D**) Immunogold electron microscopy of the thawed cryosections reveals mV-mem-opto-FGFR1 in the plasma membrane (arrowheads in **D1** and **D2**) and in the limiting membrane as well as inside various endocytic compartments (arrows in **D2** and **D3**); MVB = multivesicular body, LE = late endosome, LY = lysosome.

**Figure 3 cells-08-00243-f003:**
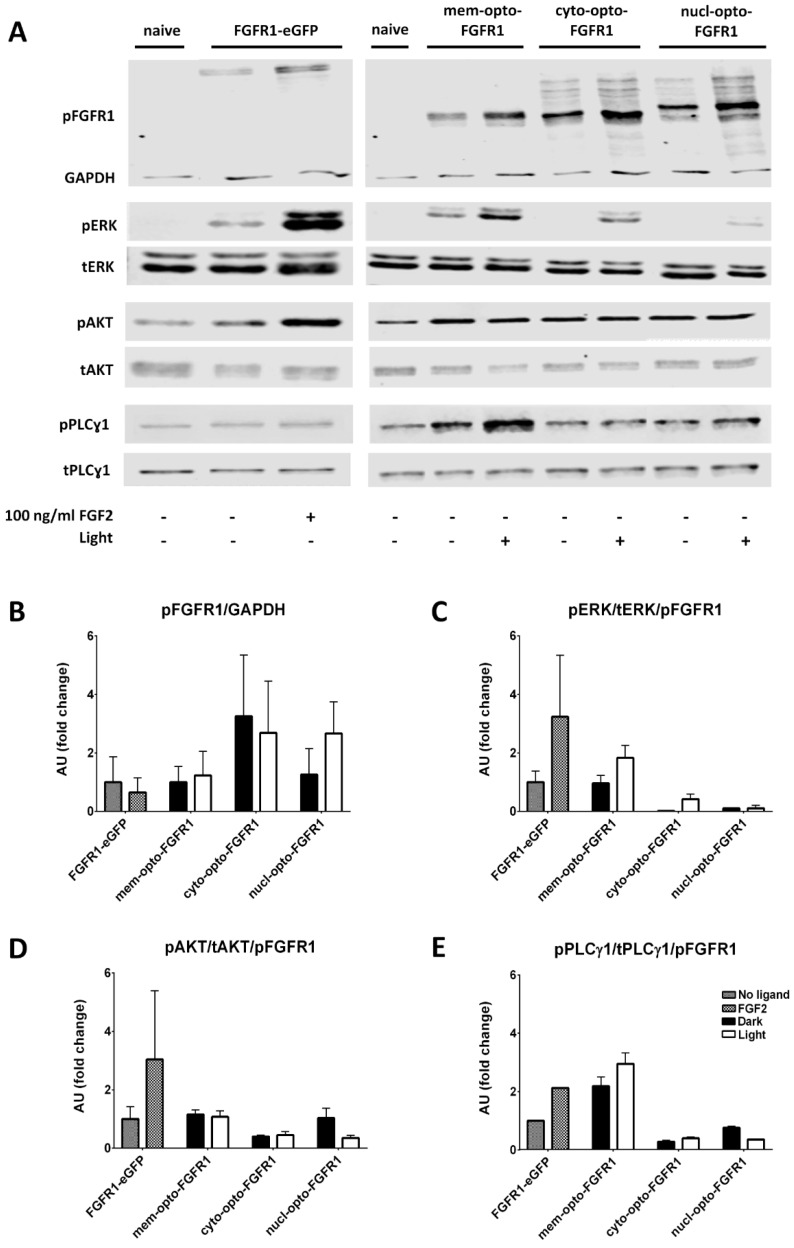
Immunoblot analysis of light- and FGF2-induced activation of key signaling molecules in HEK293 cells. (**A**) Representative examples of Western blots for pFGFR1/GAPDH, pERK/tERK, pAKT/tAKT and pPLCɣ1/tPLCɣ1 are shown. (**B**–**E**) Average intensities of ERK, AKT, and PLCɣ1 phosphorylation were quantified after normalization for pFGFR1/GAPDH to control for differences in plasmid expression levels. All ratios are relative to FGFR1–eGFP (=1) and calculated from three independent experiments (mean ± SEM).

**Figure 4 cells-08-00243-f004:**
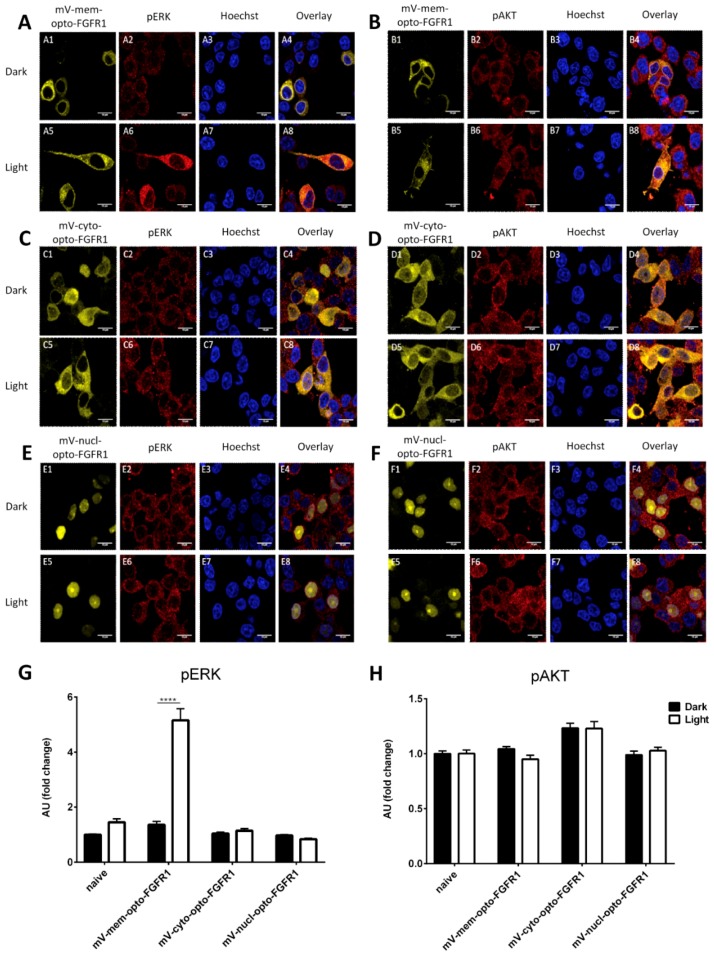
Light-induced ERK and AKT activation in mV-opto-FGFR1-transfected PC12 cells. Although mV-mem-opto-FGFR1-transfected cells exhibit short neurite extensions (sprouts) in the dark (A1), blue light induces neuronal differentiation with long, slender neurites in transfected cells (A5). (**A**) Significantly increased cytoplasmic pERK levels following light stimulation of mV-mem-opto-FGFR1-transfected cells, while non-transfected cells show no changes. (**C**,**E**) The pERK level is unchanged after stimulation of mV-cyto- and nucl-opto-FGFR1-transfected cells. (**B**,**D**,**F**) The fluorescence intensities of pAKT signals are similar in mV-opto-FGFR1-transfected and non-transfected cells in the dark and no changes are observed after blue light stimulation. Images are acquired using the same laser intensity for both dark and light in each fluorescent channel and presented without adjusting contrast or subtracting background. (**G**,**H**) Quantification of average fluorescence intensities. Results are calculated from two independent experiments and presented as mean ± SEM (30 < n < 60), **** *p* < 0.0001. Scale bars = 10 μm.

**Figure 5 cells-08-00243-f005:**
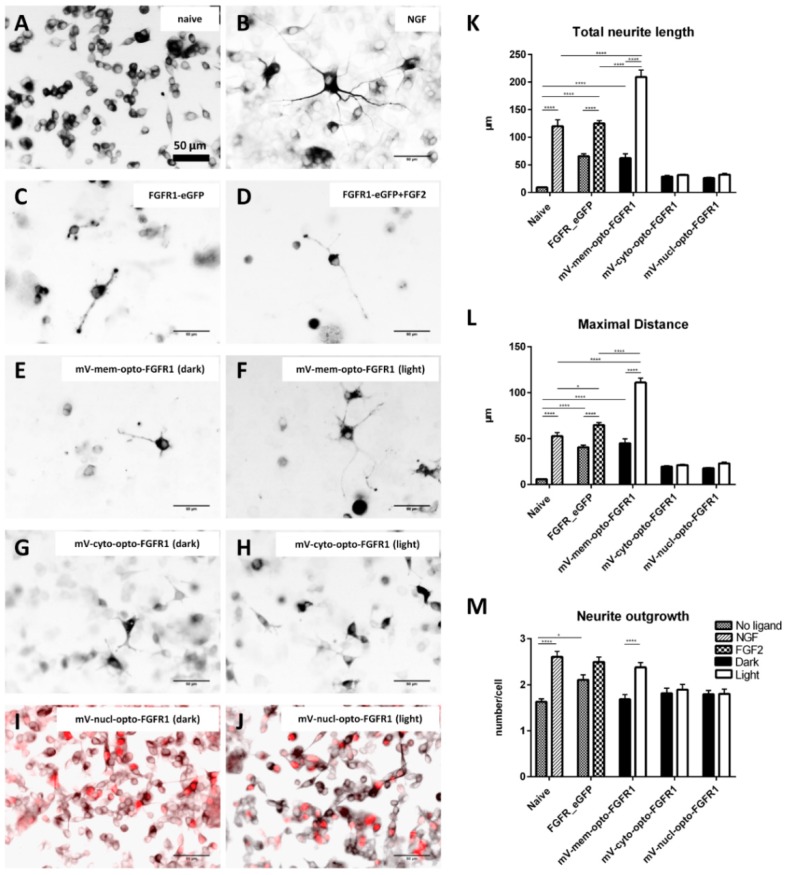
Ligand- and light-induced neurite outgrowth by pheochromocytoma (PC12) cells. (**A**–**J**) Inverted immunofluorescence images following neuron-specific class III β-tubulin staining to identify neurites (red nuclei in nucl-opto-FGFR1 cells allow identification of transfected cells in **I**/**J**). (**K**–**M**) Quantification of morphological parameters (total neurite outgrowth, longest process and number of processes per cell; see Appendix A for details). Results are calculated from three independent experiments and presented as mean ± SEM (50 < n < 100), * *p* < 0.05, **** *p* < 0.0001. Scale bars = 50 μm.

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
