# Peer review of "Membrane-Associated, Not Cytoplasmic or Nuclear, FGFR1 Induces Neuronal Differentiation"

_cells, 2019, doi:10.3390/cells8030243_

Round 1

Reviewer 1 Report

In this manuscript, Csanaky et al., investigate how the cellular localization of the FGFR1 kinase domain influences signalling and neurite outgrowth. In particular, the authors compare the activity of membrane located, cytosolic and nuclear FGFR1 kinase domain.  This comparison is made possible by fusion of the FGFR1 kinase domain to a localization domain, in addition to a LOV domain for light-induced dimerization and activation. Investigation of FGFR1 signalling from different cellular locations, including the nucleus, is particularly interesting since the receptor has been reported to enter the nucleus. It is in many cases not clear what the role of the nuclear receptor is.

First, the authors investigate the ability of their constructs to activate signalling pathways in HEK cells and PC-12 cells. Next, the authors examine the effects of their constructs on neurite outgrowth. The manuscript is well written, and the authors describe their findings concisely. However, the abstract and the discussion part of the manuscript could be more precise.

Comments:

1)      Due to low transfection efficiency in PC-12 cells, the authors investigate signalling in PC-12 cells by immunocytochemistry using p-ERK and p-AKT antibodies (Figure 4). The authors do not detect activation of AKT signalling, neither by the membrane, cytoplasmic nor the nuclear kinase. A positive and a negative control for the pAKT staining are necessary to include, in order to validate the result.

2)      Why is nuclear kinase signalling not quantified in Figure 4?

3)      Could the authors comment on why the pERK staining is generally cytosolic? How specific is the antibody?

4)      Figure labelling A-F is missing in Figure 4.

5)      Figure 4 would be easier to assess if the images were larger. (Perhaps, Hoechst staining alone is not necessary to include?)

6)      The abstract could be more precise. For example, Line 19: “Activation of the cytoplasmic FGFR1 kinase enhanced ERK signalling but did not induce neuronal differentiation.” In Figure 3, it is clear that the cytoplasmic kinase activates ERK signalling in HEK cells, while in PC-12 there is no activation of ERK signalling (Figure 4). The neurite outgrowth is investigated in PC-12 cells.  Therefore, the sentence starting at Line 19 is somewhat misleading. The authors need to take care, the abstract should be rewritten and the content be made more precise.

7)      In line 259 the authors write that: “PC-12 cells exhibited no spontaneous or FGF2 induced neurite outgrowth suggesting that the clone used in the present study does not express significant levels of endogenous FGF receptors…” The data showing FGF2 stimulation of PC-12 cells is not included in the manuscript. This should be included, or the sentence must be corrected.

8)      In line 354: “Nevertheless, the activated kinase domain alone does not appear to be sufficient to modify gene expression.” The authors have only investigated signalling and neurite outgrowth, so it would be more correct to write “…sufficient to modify neurite outgrowth.”

Author Response

The authors sincerely thank all 3 reviewers for their constructive comments regarding our Ms on optogenetic activation of FGF receptor type 1 in two different cell lines (HEK, PC12). We carefully considered their criticism, performed additional experiments and hope to have sufficiently addressed all points as follows: 

Reviewer 1

1)     Due to low transfection efficiency in PC-12 cells, the authors investigate signalling in PC-12 cells by immunocytochemistry using p-ERK and p-AKT antibodies (Figure 4). The authors do not detect activation of AKT signalling, neither by the membrane, cytoplasmic nor the nuclear kinase. A positive and a negative control for the pAKT staining are necessary to include, in order to validate the result.

Control stainings were performed with antibodies against pAKT (#4060, 1:400, Cell Signaling Technology). Images were taken by confocal microscopy (Leica SP8) at 40x magnification with identical laser settings (please see images in pdf enclosed). PC12 cells were serum starved for 2 hours before NGF treatment (100 ng/ml). These experiments are now mentioned on page 7, 3rd para.

Moreover, other groups provided immunocytochemical specificity controls for the same pAKT antibody that was used in our studies:

1. Zhang K, Duan L, Ong Q, Lin Z, Varman PM, Sung K, Cui B. Light-mediated kinetic control reveals the temporal effect of the Raf/MEK/ERK pathway in PC12 cell neurite outgrowth. PLoS One. 2014 Mar 25;9(3):e92917. doi: 10.1371/journal.pone.0092917. eCollection 2014.

2. Molgaard S, Ulrichsen M, Olsen D, Glerup S. Detection of phosphorylated Akt and MAPK in cell culture assays. MethodsX. 2016 Apr 29;3:386-98. doi: 10.1016/j.mex.2016.04.009. eCollection 2016.

2)      Why is nuclear kinase signalling not quantified in Figure 4?

Quantification of ERK and AKT activities after transfection of nuclear FGFR1 kinase has been added to Figure 4 (G and H). The authors apologize for this oversight.

3)      Could the authors comment on why the pERK staining is generally cytosolic? How specific is the antibody?

Our immunocytochemical results reveal pERK immunolabeling mainly in the cytoplasm, but only weak fluorescence in the nucleus 55 minutes after blue light stimulation. In contrast to other cell lines, activated ERK variants (ERK2–MEK1 fusion protein) diffusely localize to the cytoplasm and to extending cellular processes. This pattern suggests that pERK is involved in promoting neurite formation not only through its actions on gene transcription but also through local effects in neurites (discussed on page 11, 3rd para). Moreover, we regularly observe in PC12 cells and primary neurons higher ERK levels in the cytoplasm than in the nucleus. References:

1. Robinson, M. J., Stippec, S. A., Goldsmith, E., White, M. A. & Cobb, M. H. A constitutively active and nuclear form of the MAP kinase ERK2 is sufficient for neurite outgrowth and cell transformation. Curr. Biol. 8, 1141-1150 (1998)

2. Marvaldi, L. et al. Enhanced axon outgrowth and improved long-distance axon regeneration in

Sprouty2 deficient mice. Dev. Neurobiol. 75, 217-231 (2015).

4)      Figure labelling A-F is missing in Figure 4.

Corrected.

5)      Figure 4 would be easier to assess if the images were larger. (Perhaps, Hoechst staining alone is not necessary to include?)

Brightness and contrast of all images in Fig. 4 were equally enhanced to denote the key characteristics of stimulated PC12 cells. Removal of Hoechst images would not significantly enlarge cellular details in the other images. 

6)      The abstract could be more precise. For example, Line 19: “Activation of the cytoplasmic FGFR1 kinase enhanced ERK signalling but did not induce neuronal differentiation.” In Figure 3, it is clear that the cytoplasmic kinase activates ERK signalling in HEK cells, while in PC-12 there is no activation of ERK signalling (Figure 4). The neurite outgrowth is investigated in PC-12 cells.  Therefore, the sentence starting at Line 19 is somewhat misleading. The authors need to take care, the abstract should be rewritten and the content be made more precise.

The reviewer is correct, so the statement in the Abstract had to be re-phrased (starting at the 9th line of the Abstract). The cytoplasmic FGFR1 kinase-induced ERK signaling was enhanced in HEK293 cells, while in PC12 cells no ERK activation was measured with consequential lack of neuronal differentiation.

7)      In line 259 the authors write that: “PC-12 cells exhibited no spontaneous or FGF2 induced neurite outgrowth suggesting that the clone used in the present study does not express significant levels of endogenous FGF receptors…” The data showing FGF2 stimulation of PC-12 cells is not included in the manuscript. This should be included, or the sentence must be corrected.

Additional experiments were performed. These included ligand (FGF2 and NGF) induced neurite growth by the PC12 cell clone used in the present study and determination of endogenous FGFR levels. Please see supplementary Fig. S5 and legend for details.

8)      In line 354: “Nevertheless, the activated kinase domain alone does not appear to be sufficient to modify gene expression.” The authors have only investigated signalling and neurite outgrowth, so it would be more correct to write “…sufficient to modify neurite outgrowth.”

We agree with the reviewer. The first sentence on page 12 was changed accordingly.

Reviewer 2 Report

In their manuscript, the authors have described, "Membrane-associated but not cytoplasmic or nuclear 2 FGFR1 induces neuronal differentiation".

Comments

The authors used HEK293 and PC12 cells to investigate the effects of optogenetic stimulation of fibroblast growth factor receptors (FGFR1) on neurite outgrowth. The authors conclude that only membrane bound opto-FGFR1 constructs are capable of activating the ERK pathway sufficiently to induce neuronal cell differentiation in PC12 cells. This study is interesting and provide valuable information.

Major comments

The major concern of this study is that the authors missed a crucial control using ERK inhibitor in order to determine the functional relevance of ERK activity in the FGFR1 stimulation with a selective inhibitor (U0126) of the MEK1/2 kinases responsible for phosphorylation of ERK and consequently inhibit neuronal cell differentiation in PC12 cells which would support the specificity of the effect of Blue light stimulation and finally their conclusion. 

Figure legends: the explanations in the figure 3 and 5 are not enough and clear, therefore, the authors should add more details.

Author Response

The authors sincerely thank all 3 reviewers for their constructive comments regarding our Ms on optogenetic activation of FGF receptor type 1 in two different cell lines (HEK, PC12). We carefully considered their criticism, performed additional experiments and hope to have sufficiently addressed all points as follows: 

Reviewer 2

1. The major concern of this study is that the authors missed a crucial control using ERK inhibitor in order to determine the functional relevance of ERK activity in the FGFR1 stimulation with a selective inhibitor (U0126) of the MEK1/2 kinases responsible for phosphorylation of ERK and consequently inhibit neuronal cell differentiation in PC12 cells which would support the specificity of the effect of Blue light stimulation and finally their conclusion.

The reviewer correctly asks for a crucial control experiment which we performed. The involvement of MEK/ERK in blue light induced ERK stimulation and neuronal differentiation is now mentioned in the final para of the Results (page 9). Corresponding data are shown in Fig. S6 (legend on page 15).

2. Figure legends: the explanations in the figure 3 and 5 are not enough and clear, therefore, the authors should add more details.

Both legends have been modified (please see pages 13-14).

Reviewer 3 Report

The article of K. Csanaky and co-authors is devoted to the investigation of the activity of different chimeric forms of FGF receptors via its relation to neuronal differentiation and neurites outgrowth. The article allows to take a fresh look on the functional dependence of subcellular localization FGF receptors in cell membrane, cytoplasm or nucleus which is very interesting. The data obtained in this paper undoubtedly deserve close attention and can change our understanding of the mechanism of neuronal differentiation. Earlier it was shown that FGF receptors naturally occurs in all these cell compartments. Authors found that only membrane-bound fraction is working in neuronal differentiation, like induced by NGF. Moreover, authors observed increased level of pERK and pPLCɣ1 but not pAKT.

Obtained data supports the hypothesis proposed in the article on the role of membrane-bound fraction of FGF-receptors in neuronal differentiation.  Authors note that transcription is not the main regulation level. Although at the same time ERK can be translocated to the nucleus after phosphorylation, where it leads to changes in gene expression by phosphorylating various transcription factors. In general, the work makes a very good impression, but I have some specific comments.

1. On page 11, Figure 5 Scale bar is too small. It is also not entirely clear from the figure how authors counted a number of neurites. A simple scheme showing the counting method will be useful.

2. It is known that MEK-ERK and PI3K-AKT signaling pathways regulate neurite elongation, branching and axonal outgrowth. Authors concluded that membrane-bound FGFR, in contrast to TrkA, causes a significantly stronger effect on the pERK than on the AKT pathway. Can authors speculate about this, for example, the participation of pERK in axonal outgrowth or poorly branched dendrites because FGF PI3K-AKT signaling pathway is not activated?

3) Does it mean that the difference between function of membrane-bound and cytoplasmic FGFR includes additional factors near the cell membrane like scaffold proteins or, probably a Par3/Par6/aPKC signaling pathway is also involved in this process?  

Author Response

The authors sincerely thank all 3 reviewers for their constructive comments regarding our Ms on optogenetic activation of FGF receptor type 1 in two different cell lines (HEK, PC12). We carefully considered their criticism, performed additional experiments and hope to have sufficiently addressed all points as follows: 

Reviewer 3

1. On page 11, Figure 5 Scale bar is too small. It is also not entirely clear from the figure how authors counted a number of neurites. A simple scheme showing the counting method will be useful.

The scale bar was magnified (please see Fig. 5A). All scale bars (A-J) are of the same length. The legend was modified accordingly. A schematic describing morphometric measurements of differentiated PC12 cells is provided in Fig. S1 (legend on page 14).

2. It is known that MEK-ERK and PI3K-AKT signaling pathways regulate neurite elongation, branching and axonal outgrowth. Authors concluded that membrane-bound FGFR, in contrast to TrkA, causes a significantly stronger effect on the pERK than on the AKT pathway. Can authors speculate about this, for example, the participation of pERK in axonal outgrowth or poorly branched dendrites because FGF PI3K-AKT signaling pathway is not activated?

The reviewer mentions a very important point that we discussed in the final part of the Discussion (page 11, 3rd para). A variety of studies applying selective ERK activation demonstrated that Raf/MEK/ERK signaling is primarily responsible for neurite extension. Please see Hausott and Klimaschewski (2016) for details (Hausott B, Klimaschewski L. 2016. Membrane turnover and receptor trafficking in regenerating axons. Eur. J. Neurosci. 43, 309-317).  

3. Does it mean that the difference between function of membrane-bound and cytoplasmic FGFR includes additional factors near the cell membrane like scaffold proteins or, probably a Par3/Par6/aPKC signaling pathway is also involved in this process?  

This is a very interesting point. We assume that functional signaling platforms inducing a variety of pathways (ERK, AKT, PLC/PKC) form in association with membranes only. At plasma membranes active RTKs can recruit different adaptors and scaffold proteins as compared to other cellular membranes which exhibit different curvatures and different phospho-inositol compositions. We added these aspects to the final para of the Discussion.

Round 2

Reviewer 1 Report

The authors have answered my previous comments in a satisfactory way.

Reviewer 2 Report

The ppaer is accepted in this form